



# Flow rate and source reservoir identification from airborne chemical sampling of the uncontrolled Elgin platform gas release.

James D. Lee[1*], Stephen D. Mobbs[2], Axel Wellpott[3], Grant Allen[4], Stephane. J-B. Bauguitte[5], Ralph R. Burton[2], Richard Camilli[5], Hugh Coe[4], Rebecca E. Fisher[6], James L. France[6,7], Martin Gallagher[4], James R. Hopkins[1], Mathias Lanoiselle[6], Alastair C. Lewis[1], David Lowry[6], Euan G. Nisbet[6], Ruth M. Purvis[1], Sebastian O'Shea[3], John A. Pyle[8], Thomas B. Ryerson[9].

[1] National Centre for Atmospheric Science, University of York, York, UK.
[2] National Centre for Atmospheric Science, University of Leeds, Leeds, UK.
[3] Facility for Airborne Atmospheric Measurements, Cranfield University, Bedford, UK.
[4] School of Earth, Atmospheric and Environmental Sciences, University of Manchester, Manchester, UK,
[5] Department of Applied Ocean Physics and Engineering, Woods Hole Oceanographic Institution, Woods Hole, MA, USA.
[6] Department of Earth Sciences, Royal Holloway University of London, Egham, TW20 0EX, UK.
[7] School of Environmental Sciences, University of East Anglia, Norwich, UK.
[8] National centre for Atmospheric Science, Department of Chemistry, University of Cambridge, Cambridge, UK
[9] Chemical Sciences Division, Earth System Research Laboratory, NOAA, Boulder, CO, USA.

*Corresponding author, email: james.lee@york.ac.uk*

**Abstract.** An uncontrolled gas leak from 25 March to 16 May 2012 led to evacuation of the Total *Elgin* well head and neighbouring drilling and production platforms in the UK North Sea. Initially the atmospheric flow rate of leaking gas and condensate was very poorly known, hampering environmental assessment and well control efforts. Six flights by the UK FAAM chemically-instrumented BAe-146 research aircraft, were used to quantify the flow rate. Where appropriate, two different methods were used to calculate the flow rate: 1. Gaussian plume fitting in the vertical and 2. Direct integration of the plume. When both methods were used, they compared within 6% of each other and within combined errors. Data from the first flight on 30 March 2012 showed the flow rate to be $1.3\pm0.2$ kg $CH_4$ s$^{-1}$, decreasing to less than half that by the second flight on 17 April 2012. $\delta^{13}C_{CH4}$ in the gas was found to be –43 ‰, implying that the gas source was unlikely to be from the main high-pressure high-temperature Elgin gas field at 5.5 km depth, but more probably from the overlying Hod Formation at 4.2 km depth. This was deemed to be smaller and more manageable than the high-pressure Elgin field and hence the response strategy was considerably simpler. The first flight was conducted within 5 days of the blowout and allowed a flow rate estimate within 48 hours of sampling, with $\delta^{13}C_{CH4}$ characterisation soon thereafter, demonstrating the potential for a rapid-response capability that is widely applicable to future atmospheric emissions of environmental concern. Knowledge of the *Elgin* flow rate helped inform subsequent decision making. This study shows that leak assessment using appropriately designed airborne plume sampling strategies is well suited for circumstances where direct access is difficult or potentially dangerous. Measurements such as this also permit unbiased regulatory assessment of potential impact, independent of the emitting party, on timescales that can inform industry decision-makers and assist rapid response-planning by government.



## 1 Introduction

Elgin is a high pressure/high temperature methane and condensate field in the Central Graben of the UK North Sea, about 240 km East of Aberdeen, set in 93 m of water (Isaksen, 2004) (see figure 1). On 25 March 2012, an accidental and uncontrolled hydrocarbon release occurred at the *22/30c*-G4 well, which penetrates the Elgin reservoir at a depth of approximately 5.5 km. This led to the abandonment of the Elgin platform and evacuation of non-essential personnel from nearby facilities. Actions taken in response to this incident shut down or

affected nearly 10% of the UK natural gas supply for 6-7 weeks. The well was eventually capped on 16 May 2012.

The Elgin gas well was known to produce both natural gas (mainly methane) and natural gas condensate (Fort and Senequier, 2003). The presence of condensate and gas led to additional concerns regarding a potential

fuel/air explosion. The resulting abandoning of the platform meant that quantification of the gas emission was challenging. The $H_2S$ concentrations in the main field (~45 ppm) are close to what is generally considered safe exposure limits ((US), 2009), so conventional response assessment operations would require additional human health and safety precautions. As a result, remote methods were sought and an aerial survey, due to the fact that it would limit the duration and concentration of human exposure to the plume, was deemed appropriate. In

response, within five days of abandoning the platform, the Natural Environment Research Council / UK Met Office Facility for Airborne Atmospheric Measurements (FAAM) deployed its chemically-instrumented BAe-146 research aircraft to measure the gas plume from the release and to take whole-air samples of the air for subsequent laboratory characterisation. The aircraft was equipped with a range of instruments including continuous methane measurement by cavity-enhanced absorption spectroscopy (Fast Greenhouse Gas Analyzer,

Los Gatos Research Inc). Whole air grab sampling was carried out by two independent systems: the aircraft's in-built stainless steel flasks sampling facility and also manually into 3l Tedlar bags. Data from 6 flights from 30[th] March to 15[th] August 2012 are available and presented below. The aircraft data were used to successfully characterise the leaking gas (flow rate and composition), allowing a plan for remedial action at the well head to be implemented. This paper presents the analysis of these data.


## 2 Experimental

### 2.1 Instrumented research aircraft


The FAAM aircraft manages a modified BAe-146-300 aircraft which carries core and optional instruments for measuring various components of the atmosphere. Core instruments cover a range of basic atmospheric measurements including thermodynamic properties, wind, turbulence and some chemical species. These are provided by FAAM as part of the facility. Details of most FAAM instruments can be found on the FAAM web-

site: http://www.faam.ac.uk. Wind and turbulence are measured using a five-port pressure measurement system in the aircraft radome, combined with two scientific static ports located symmetrically on either side of the



aircraft. Wind and thermodynamic profiles from the aircraft down to the surface are also provided by dropsondes which can be released and tracked periodically in flight. Of greatest relevance to the work reported here are the systems for fast methane measurement and for obtaining air samples for laboratory analysis. These are described below.

## 2.2 Atmospheric measurements

$CO_2$ and $CH_4$ were measured in situ on the aircraft using a modified Los Gatos Research Inc. Off-Axis Integrated Cavity Output Fast Greenhouse Gas Analyser (FGGA model RMT-200). This was calibrated in-flight against gas standards certified by the Max-Planck Institute for Biogeochemistry (Jena) as part of the Infrastructure for Measurements of the European Carbon Cycle project (EU 13 IMECC; see http://imecc.ipsl.jussieu.fr/). The stability of these standards was also cross-checked against Royal Holloway laboratory standards. All reported $CH_4$ mixing ratio data are traceable to the National Oceanographic and

Atmospheric Administration NOAA-04 scale (Dlugokencky et al., 2005). A technical summary of the FGGA deployed on-board the FAAM aircraft, the calibration system, data analysis and quality control methods developed by the University of Manchester and FAAM is presented elsewhere (O'Shea et al., 2013), illustrating the airborne performance of the system, chiefly a measurement accuracy of $\pm1.28$ppb with a $1\sigma$ precision at 1 Hz of 2.48 ppb for $CH_4$.

Ambient air was sampled using both the automated whole air sampling (WAS) system fitted to the aircraft and manually into Tedlar bags for post-flight laboratory analysis. The WAS system consists of sixty-four silica passivated stainless steel canisters of three litre internal volume (Thames Restek, Saunderton UK) fitted in packs of 8, 9 and 15 canisters to the rear lower cargo hold of the aircraft. Each pack of canisters was connected to a

3/8 inch outside diameter stainless steel sample line, in turn connected to an all-stainless steel assembly double-headed three phase 400 Hz metal bellows pump (Senior Aerospace, USA). The pump drew air from the portside ram air sample pipe and pressurized air into individual canisters to a maximum pressure of 3.25 bar, giving a useable sample volume for analysis of up to 9 litres. WAS canisters take approximately 20 seconds to fill at typical boundary layer pressures, and so they provide an averaged measure of hydrocarbon content. At a

typical aircraft science speed of around 100 m s$^{-1}$, a WAS sample is therefore an average mixing ratio over a spatial extent of ~2 km. The length of sampling manifold within the aircraft creates a delay of around 10 seconds between air entering the inlet at the front of the aircraft and being available for capture in the hold. This slight delay allowed the real-time $CH_4$ outputs from the FGGA to be used to aid the capture plume samples with canisters. The integrated nature of the WAS means that the concentrations reported do not represent peak plume

concentrations, however these can be inferred assuming a constant relationship to $CH_4$. The manual Tedlar bag sampling system employed a Metal Bellows pump (model MB-158) and was more direct, with a few seconds lag time and rapid bag filling (~ 5 secs).

Air samples were analysed for Volatile Organic Compounds (VOCs) within 48 hours of collection at the

University of York using a dual channel gas chromatograph with two flame ionisation detectors (Hopkins et al.,



2011).  One litre samples of air were withdrawn from the sample canisters and dried using a glass condensation finger held at –30°C. $C_2$-$C_7$ samples were pre-concentrated onto a multi-bed carbon adsorbent trap, consisting of Carboxen 1000 and Carbotrap B (Supelco), held at –20°C and then heated to 325°C at 16°C s$^{-1}$ and transferred to the GC columns in a stream of helium.  The eluent was split in an approximately 50:50 ratio between an

aluminium oxide ($Al_2O_3$, $NaSO_4$ deactivated) porous layer open tubular PLOT column (50 m, 0.53 um id) for analysis of NMHCs and two LOWOX columns (10 m, 0.53 um id) in series for analysis of polar VOCs. Both columns were supplied by Varian, Netherlands. Peak identification and calibration was made by reference to a part per billion level certified gas standard (National Physical Laboratory, ozone precursors mixture, cylinder number: D64 1613) for NMHCs.  This standard and instrument has in turn been evaluated as part of the WMO

GAW programme and was within target operating limits.

Methane isotopic composition ($\delta^{13}C_{CH4}$) was measured at Royal Holloway, University of London (RHUL) in samples collected in WAS canisters during flights on 30$^{th}$ March and 3$^{rd}$ April and in Tedlar bag samples collected manually on the 3 April flight.  Prior to isotopic analysis, the methane mixing ratio in the samples was

measured using a Picarro 1301 cavity ringdown spectrometer, calibrated using NOAA air standards. Repeatability in $CH_4$ mixing ratio measurements was +/- 0.3 ppb.  $\delta^{13}C_{CH4}$ was analysed using a modified gas chromatography isotope ratio mass spectrometry (GC-IRMS) system.  The methodology is described in detail by (Fisher et al., 2006).  $\delta^{13}C_{CH4}$ repeatability was ~ 0.05‰.  All isotope measurements were made in triplicate. Isotope ratios are given in $\delta$ notation on the VPDB (Vienna Pee Dee Belemnite) scale.  Keeling plot

methodology is described by (Pataki et al., 2003) and (Fisher et al., 2017).

The FAAM aircraft is equipped with a system to drop radiosondes (VAISALA, Finland). The sondes (RD94) descend on a parachute with a speed of ~10 m s$^{-1}$ and measure air pressure, air temperature, relative humidity, and GPS position on their way to the surface. Wind speed and wind direction are calculated from the GPS

measurements and the known drag of the dropsonde (Wang, 2005). Data can be received and viewed in real time on the aircraft.

### 2.3 Flight planning and safety case


The location of the gas source relative to the sea surface and the mass flux of the emission were initially not well known.  A prospective analysis of the gas plume was obtained using HYSPLIT model simulations (Stein et al., 2015), carried out using meteorological fields from the US National Centre for Environmental Prediction Global Forecast System (NCEP-GFS) (ftp://arlftp.arlhq.noaa.gov/pub/archives/gdas0p5/ref), obtained via the Air

Resources Laboratory of the National Oceanic and Atmospheric Administration. NCEP GFS data are high resolution (0.5 degree latitude and longitude and 3 hours temporally).  Figure 2 shows the modelled $CH_4$ concentration from 0-1000m above sea level, for 12.00 UTC on 2 April 2012.  The modelled start of release was 0000 UTC and the modelled release rate was 23.5 kg s$^{-1}$. The model outputs were used for flight planning and to provide a safety case for the flights. Given the explosion risk, and because hydrogen sulfide ($H_2S$) in the Elgin

reservoir was reported to be ~45 ppm (Fort and Senequier, 2003), close to the safe human exposure limit, a risk





reduction analysis was carried out prior to the first BAe-146 research flight to specify the "turn away" concentrations based on real-time measurements on-board the aircraft using hand held sensors. The flights did not enter a 3 nautical mile radius, 4000 ft altitude exclusion zone imposed by the UK Maritime and Coastguard Agency at the time of the emergency. Outside of this excluded volume, a "turn away" detection value of 40

ppm $CH_4$ was established, which was 20 times the background concentration, 10 times higher than the forecast of $CH_4$ likely to be present (given an unrealistically high leak rate of 23.5 kg s$^{-1}$ set in the model) and 100 times below any possibly dangerously combustible concentration of the worst case gas mixture.

**3 Flow rate calculation**

The plume of $CH_4$ and other gases was assumed to be neutrally buoyant and non-reacting (on the time and distance scales involved in the aircraft measurements). The fundamental assumption is that the plume dispersion may be modelled by a Gaussian distribution (see, e.g., Turner 1994):

$$C(x, y, z) = \frac{q}{\pi \sigma_y \sigma_z U} \exp\left( -\frac{(y - y_0)^2}{2\sigma_y^2} - \frac{z^2}{2\sigma_z^2} \right)$$     (Eq 1)

where $q$ is the source strength (mass emission rate) of the methane leak, $C(x,y,z)$ is the molar concentration which varies in the $x$ (downwind), $y$ (cross-wind) and $z$ (vertical) directions and $U$ is the mean prevailing wind speed. The $\sigma^2_y$ and $\sigma^2_z$ terms are the mean squared distances of the plume spread in the cross-wind and vertical directions (both growing by dispersion with down-wind distance). The source is fixed at $x = 0$ and $z = 0$. The reason for not taking the centre-line of the plume to necessarily be at $y = 0$ is that during cross-plume aircraft

flights, the cross-wind position, $y_0$ of the plume was determined directly from the measurements for each pass. The assumptions (and rationale) underlying Eq. (1) are:

1. The mean prevailing wind velocity does not exhibit strong shear in the vertical or significant variability over the course of the sampling. This includes both changes in speed U and direction and was confirmed by measured

wind data.
2. The height above the sea surface of the source may be neglected. Although relatively straightforward to include, other uncertainties in the calculations of the flow rate make this parameter negligible.
3. Similarly, any effect on the turbulent vertical mixing of structural down-wash from the rig structure is not detectable (i.e. the plume is seen to be well-mixed in down-wind sampling).

4. There is negligible vertical restriction of dispersion by capping inversions or the boundary-layer top (as the plume was not observed to rise to the local MBL top at the point of aircraft sampling).

Assumption (4) is clearly not always valid. It is relatively straightforward, from a theoretical point of view, to account for a restricted mixing height $H$:



$$C(x, y, z) = \frac{q}{\pi\sigma_y\sigma_z U}\exp\left(-\frac{(y-y_0)^2}{2\sigma_y{}^2}\left[\exp\left(\frac{z^2}{2\sigma_z{}^2}\right) + \exp\left(-\frac{(z+2H)^2}{2\sigma_z{}^2}\right) + \exp\left(-\frac{(z-2H)^2}{2\sigma_z{}^2}\right)\right]\right)$$

(Eq 2)

However, fitting of Eq. (2) to experimental data with large uncertainties is not feasible. Mathematically, fitting is relatively straightforward but in practice it is not possible to distinguish reliably between the effect of an elevated inversion and a general reduction in vertical spreading $\sigma_z$.

Far down-wind, in the presence of an elevated inversion which strongly inhibits mixing above height $H$, the

pollutant is thoroughly mixed below the inversion and further mixing results only in horizontal spreading. Then a much simpler Gaussian plume model may be used (Ryerson et al., 2011):

$$C(x, y) = \frac{q}{\sqrt{2\pi}\sigma_y UH}\exp\left(-\frac{\left(y-y_0\right)^2}{2\sigma_y{}^2}\right)$$

(Eq 3)

Based on the above theoretical considerations, a sampling strategy was used which follows closely that used by (Ryerson et al., 2011) during the 2010 Deepwater Horizon spill in the Gulf of Mexico. The basis of the method is to sample the cross-wind structure of the plume using repeated aircraft passes across the plume down-wind of the source. The repeated cross-plume sampling aims to determine the cross-wind structure ($\sigma_y$, $y_0$) and peak concentration, plus to determine how these parameters vary in the vertical and in the down-wind direction.

Sampling across the plume was carried out at different altitudes within the marine boundary layer to assess the vertical dispersion of the plume, which is required by both analysis methods which we now describe. Two different analysis approaches have been used, determined by the outcome of these measurements. They are referred to as Method 1 and Method 2 in this manuscript.

*Method 1: Gaussian fitting in the vertical*

Method 1 is appropriate when there exists no significant temperature inversions at levels where $z \leq \sigma_z$. This requires that measurements are made up to a height of at least $\sigma_z$ and that no inversions are encountered up to that level. If an inversion layer does exist, then method 1 may still be used if the measured value of $\sigma_z$ is such that $\sigma_z << H$, where $H$ is the mixing layer height.


Writing Eq. (1) as:





$$C(x, y) = C_0 \exp\left(-\frac{(y - y_0)^2}{2\sigma_y{}^2}\right) \qquad \text{(Eq 4)}$$

where:

$$C_z(x, z) = C_0(x)\exp\left(-\frac{z2}{2\sigma_z{}^2}\right) \qquad \text{(Eq 5)}$$

and:

$$C_0 = \frac{q}{\pi\sigma_y\sigma_z U'} \qquad \text{(Eq 6)}$$

then $C_z$ and $\sigma_z$ may be obtained from fitting cross-plume data at fixed distance down-wind to Eq. (6). Then
20    writing Eq. (5) in the form:

25    $$\ln(Cz) = \ln(C_0) - \frac{z^2}{2\sigma_z{}^2} \qquad \text{(Eq 7)}$$

$C_0$ and $\sigma_z$ can be obtained by plotting $Cz$ against $z^2$ using data from all transect levels at a fixed downwind
distance.

30

*Method 2: Fully mixed layer*




This approach is appropriate when the airborne measurements fully define the vertical extent of plume mixing (e.g., (Conley et al., 2016), or the plume is mixed thoroughly in the vertical up to a capping inversion (e.g., (Ryerson et al., 2012)), such that:

- there exists a clear temperature inversion/elevated stable layer in atmospheric profiles revealed using aircraft measured thermodynamic profiles, dropsondes or radiosondes, and
- cross-wind transects show little decrease of concentration with height (within the uncertainties), up to the inversion level.

Assuming conditions are suitable for method 2, then writing Eq. (3) as

$$C(x, y) = C_0 \exp\left(-\frac{(y - y_0)^2}{2\sigma_y^2}\right) \qquad \text{(Eq 8)}$$

where:

$\qquad C_0 = \dfrac{q}{\sqrt{2\pi}\sigma_y U H'}$ \qquad (Eq 9)

Best fitting of the concentration measurements to Eq. (8) is used to determine $C_0$, $y_0$ and $\sigma_y$ and then the leak rate $q$ is determined from Eq. (9), using estimates of the inversion height $H$ from the atmospheric soundings.

Either of these methods allow for calculation of the mean emission flow rate in a relatively short time period after measurements are taken (potentially within 24 hours). This makes airborne sampling useful for emergencies where fast quantification of flow rate can be critical for informed decision making.

**4 Results**


Flights to sample the plume emanating from the Elgin platform were carried out on 30[th] March, 3[rd] April, 17[th] April, 24[th] April, 4[th] May and 15[th] August 2012. Figure 3 shows flight tracks for all the flights, with local wind direction (as measured from the aircraft) indicated as a wind barb. The tracks show the position of the Elgin platform, along with others in the immediate area, and the legs sampling the plume at different distances from
the source.





### 4.1 CH$_4$ leak rate

Measurements of CH$_4$ were taken at different heights above sea level and different distances from the platform on each of the flights. Figure 4 shows CH$_4$ mixing ratios taken on each flight, plotted as a function of distance along the flight track perpendicular to the plume for all flights at 5 and 15 NM from the *Elgin* platform. To aid the analysis, profiles of potential temperature and wind speed and direction from dropsonde data taken early and late in the flights (where available) are shown in figure 5. Thermodynamic profiles measured by on-board instrumentation during plume sampling were found to be consistent with the dropsonde data.

*Flight B688 – 30$^{th}$ March:*
Initially two passes were made across the line of the expected plume but around 10 NM upwind of the Elgin rig. These provided background methane concentrations. The aircraft was then repositioned downwind of the rig and repeated passes were made across the plume at two distances (approximately 5 NM and 15 NM) from the rig, with mean wind speeds in the range 12-20 ms$^{-1}$ (measured from the aircraft). Measured methane concentrations across the plume at approximately 5 NM downwind show a very clearly defined plume with a peak of around 1000ppb above background at a measurement height of 35m, whilst at 15 NM the plume has become more broken and indeed for one of the passes it appears to have split into two separate plumes. At both distances from the rig, the plume peak concentration decreases with height. The decrease is evidence for the plume not being fully mixed up to an inversion level. For this flight, which was made with a short preparation period, no dropsonde was launched. We do however have available a profile from a radiosonde launched at the time of the flight from the nearby Ekofisk rig. Data from this is shown in the SOM (figure S1(a)). There is clear evidence of a temperature inversion at around 750 m. However, the fitted plume parameters suggest that mixing had not occurred up to this level, even at 15 nm downwind. Therefore method 1, the Gaussian fitting in the vertical, has been used for flow rate estimation (all flow rates results will be discussed at the end of this section). CH$_4$ flow rates of 1.10 ± 0.55 kgs$^{-1}$ and 1.06 ± 0.49 kgs$^{-1}$ were calculated using this method for the 5 NM and 15 NM passes respectively.

*Flight B689 – 3$^{rd}$ April:*
Initially two passes were made across the line of the expected plume but around 5 NM upwind of the Elgin rig, which provided background methane concentrations. The aircraft was then repositioned downwind of the rig and repeated passes were made across the plume at two distances (approximately 5 NM and 15 NM), with mean wind speeds of ~15 ms$^{-1}$ throughout. There is again evidence for a decay of peak CH$_4$ concentration with height at 5 NM downwind, consistent with the methane having not mixed through the full depth of the boundary-layer. Potential temperature profiles from dropsondes launched at the start and end of the measurement part of the flight are shown in figure 5. These show good evidence of a stable layer/inversion just above 1 km altitude early in the flight, with essentially neutrally stratified conditions present below this. These conditions persisted throughout the flight, although the later dropsonde profile shows that the stable layer above became weaker with time, likely associated with marine boundary layer heating throughout the day. The consistent decrease in plume concentration with height, coupled with the fact that the measurements were all made well below the inversion layer, suggests that the method 1, the Gaussian fitting in the vertical, can be used for flow rate calculation. At 15



NM downwind satisfactory Gaussian fits to the data are possible in all cases, however there is little evidence of a decay of concentration with height. This lack of consistent decay, plus the clear existence of an inversion layer at just above 1 km, suggests that the assumption of mixing up to the inversion height may be made here. Method 2 was therefore also used to calculate the methane flow rate from data at 15 nm and 25 nm from the rig, using a

mixing height of 1.13±0.1 km. The calculated flow rate was $0.55 \pm 0.71$ kg s$^{-1}$ using method 1 and $0.59 \pm 0.21$ kgs$^{-1}$ and $0.58 \pm 0.07$ kgs$^{-1}$ using method 2 (at 5 NM and 15 NM from the source respectively), demonstrating good agreement (within 5%) of the methods.

*Flight B690 – 17$^{th}$ April:*

Initially three passes were made across the line of the expected plume at around 5 NM upwind of the Elgin rig, which provided background methane concentrations. The aircraft was then repositioned downwind of the rig and repeated passes were made across the plume at two distances (approximately 5 NM and 20 NM), with mean wind speeds ~20 ms$^{-1}$ throughout. The  observed decay of peak concentration with height at both downwind distances is again consistent with the methane having not mixed to the top of the boundary-layer. Potential

temperature profiles from dropsondes (figure 5) launched at the start and end of the measurement part of the flight show that the atmosphere appears to be stable at all levels above a very shallow (<200 m) mixed layer close to the surface. The rather uniform stability, coupled with the decay of concentration with height, supports the use of method 1, Gaussian fitting in the vertical for flow rate calculation. The flow rate was calculated to be $0.24 \pm 0.10$ kg s$^{-1}$ and $0.45 \pm 0.31$ kg s$^{-1}$ for the 5 NM and 20 NM passes respectively.

*Flight B691 – 24$^{th}$ April:*

Initially a pass was made across the line of the expected plume at approximately 5 NM upwind of the Elgin rig. This provided background methane concentrations. The aircraft was then repositioned downwind of the rig and repeated passes were made across the plume at two distances (approximately 5 NM and 20 NM) from the rig,

with mean wind speeds $2 – 4$ ms$^{-1}$ throughout. The potential temperature profiles from dropsondes launched at the start and end of the measurement part of the flight (figure 5) show a generally stable atmosphere with some tendency to become mixed over the lowest 400 m later in the flight. There is no evidence of significant elevated inversions. At 5 NM downwind there is insufficient data for confident conclusions to be drawn, particularly because even though there is little evidence of variation of concentration with height, there is no clear mixing

height. There is evidence for a decay of peak concentration with height at 20 NM downwind, suggesting that method 1, the Gaussian fitting in the vertical, may be applied here. However, the plume transects at 5 NM show a ragged and broken plume and at 20 NM the plume is not well defined at all, behaviour that can be attributed to the very low wind speeds.  Most of the transects have produced fitted Gaussian cross-sections but these cannot be considered to be of high reliability. So although the results at 20 NM have produced a methane flow rate

using the Gaussian fitting in the vertical (method 1), there is considerable uncertainty, due to the light winds, regarding whether all of the methane plume filaments have been reliably detected and therefore reliability of the overall flow rate result must be suspect. The flow rate was calculated to be $0.06 \pm 0.29$ kg s$^{-1}$, however the principal conclusion from this flight is that stronger winds (> ~5 ms$^{-1}$) are necessary in order to reliably measure the flow rate.




*Flight B693 – 4th May:*

Once again, two passes were initially made across the line of the expected plume at around 5 NM upwind of the Elgin rig to provide the background methane concentrations. The aircraft was then repositioned downwind of the rig and repeated passes were made across the plume at two distances (approximately 5 NM and 20 NM) from the rig. At the lowest height (45 m) the $CH_4$ plume is observed to peak at ~150 ppb above background. There is evidence for a decay of peak concentration with height at both downwind distances, consistent with the methane having not mixed through the boundary-layer. The potential temperature profiles from dropsondes launched at the start and end of the measurement part of the flight (figure 5) show atmosphere to be generally stable at all levels above a shallow (<300 m) mixed layer close to the surface. There is evidence of a significant inversion above 2 km at the start of the flight but no inversion at lower levels. The data show that the methane has definitely not mixed up to 2 km. The rather uniform stability at lower levels, coupled with the decay of concentration with height, supports the use of method 1, the Gaussian fitting in the vertical, for calculating the methane flow rate. A flow rate of $0.31 \pm 0.32$ kg s$^{-1}$ was calculated for this flight.

*Flight B727 – 15th August:*

The objectives of this flight were (a) to confirm that the methane leak from Elgin had been effectively capped and (b) to gain further information concerning background sources of trace gases from oil and gas installations, in order to assist with interpretation of previous (and potential future) research flights. In support of these two aims, flight legs were made across the expected line of any plume from the Elgin rig, as in previous flights (these were made closer to Elgin than in previous flights as the air exclusion zone previously operating within 3 NM of the rig had been lifted). The primary result from this flight was that there was no detectable methane plume from the Elgin rig. The FGGA instrument is capable of resolving concentration gradients to within 2 ppb (O' Shea et al., 2013), and therefore able to discriminate emitted plumes from background variability for similar enhancements in principle. The characterisation of a limit of detection for any plume is case-study-specific as any observed enhancement must always be compared with the observed background variability, and also take into account the limitations of sampling. In the case of flight B727, we cannot make this distinction within the precision of the FGGA instrument and therefore conclude that a plume was not sampled during this flight. The potential temperature from a single dropsonde launched from close to the Elgin rig during this flight is shown in the SOM (figure S1(b)). The profile is quite unlike that observed in previous sampling, with a shallow well-mixed layer up to approximately 200 m, above which was a stable layer up to approximately 500 m. This would indicate the potential for pollutant capping below 200 m. Above 500 m the atmosphere was again well mixed. Transects were made below 200 m, between 200 and 500 m and above 500 m. In no case was an elevated methane signal above the background detected, in contrast to all previous flights. The FGGA instrument is capable of resolving concentration gradients to within 2 ppb (O' Shea et al., 2013), and therefore able to discriminate emitted plumes from background variability for similar enhancements in principle. The characterisation of a limit of detection for any plume is case-study-specific as any observed enhancement must always be compared with the observed background variability, and also take into account the limitations of sampling. In the case of flight B727, we cannot make this distinction within the precision of the FGGA instrument and therefore conclude that a plume was not sampled during this flight.



The methane flow rates calculated from the plume measurements and analysis from flights B688, B689, B690, B691 and B693 are summarised in Figure (6). Error bars have been deduced from the analysis detailed in the supplementary online material. The results indicate:

(a) There was a significant decrease in methane flow rate between 30th March and 17th April 2012, dropping from 1.08 to 0.35 kgs$^{-1}$. It worth noting that the means for the 30th March flights are outside the error bars for the 17th April flights, adding weight to the argument that the flow rate has decreased.

(b) There was no further detectable decrease in flow rate up to and including 4 May 2012.

(c) The results from the flight on 24 April 2012 are not considered trustworthy due to the extreme low wind speeds. The possibility that parts of the plume were missed due to irregular dispersion cannot be ruled and is consistent with the apparent observation that the deduced flow rate on this day was lower than any previous or subsequent day.

(d) When applicable (e.g. on flight B690) , both methods 1 and 2 described in section 3 give reliable and consistent flow rate estimates.

It is noteworthy that on only one flight (B690, 3rd April) was it possible to use the fully mixed boundary-layer assumption (method 2). This contrasts with the experience of the Deepwater Horizon incident reported by (Ryerson et al., 2012). There are several possible factors contributing to this. For the majority of the flights there was no clear capping inversion to the boundary-layer (see figure 5). Different water and air temperatures likely helped drive vertical mixing better during the Deepwater Horizon incident than the conditions present here. Although the gas temperature from the Hod formation where the gas is thought to have originated is ~165 °C, considerable cooling is likely to have occurred by the time that the gas was released into the atmosphere, due to with conductive cooling as the gas migrates up through the well and an additional temperature drop caused by pressure drop as the gas exits the leak orifice. The lower concentrations of gases from the Elgin leak required measurements to be made closer to the source than during the Deepwater Horizon incident, allowing less time for vertical mixing. The sea surface temperatures and near-surface air temperatures were similar in all cases for the Elgin flights (see figure S2 in SOM). This indicates only small air-sea heat fluxes and low tendency for buoyant generation of turbulence. All of the flights during the period of the leak indicate small sea to air heat fluxes, with this being reversed for the single August flight. This again demonstrates the importance of having the two methods for calculating the atmospheric flow rate, one of which (method 1), does not require the plume to be fully mixed in the vertical, conditions that maybe prevalent in colder environments.

**4.2 Hydrocarbon composition of the plume**





Non-methane hydrocarbons (NMHC) and other volatile organic compounds in the plumes were determined from whole-air flask samples by offline analysis (Hopkins et al., 2011;Lidster et al., 2014). NMHC content was dominated by light alkanes ranging from >20 ppb ethane to <1 ppb benzene and <0.1 ppb higher monoaromatics. A close relationship between elevated $CH_4$ and NMHCs was observed in plume samples (Figure 7) with consistent ratios. The plume was dominated by short chain ($<C_4$) linear and branched chain alkanes and some larger monoaromatic compounds, with up to five alkyl groups substituents attached to the benzene ring. No polycyclic aromatic compounds or oxygenated species were observed in any of the samples.

The spatial mixing of higher condensate species with background air was highly correlated to $CH_4$ and $C_2$-$C_4$ NMHCs, as expected from emissions from a single point source. Atmospheric measurements showed a lower proportion of $>C_6$ species than in Fort and Sénéquier (2003) for Elgin reservoir fluids (~3 % vs 13 %). We speculate that these larger species condensed as liquids to the relatively cold (7 to 8 °C) sea surface rather than being transported in the gas phase into the air plume. The corollary is that the NMHC data show no evidence for widespread higher condensate evaporation into air from the seawater sheen, despite reports of significant pollution risks, including condensates from underwater release. This would suggest condensate removal was by biological processes in the water or simply due to cold surface water decreasing the evaporation rate to undetectable levels.

NMHC analyses reported here demonstrate that potential fractionation may have occurred as the gas/liquid mix was emitted from the leak, and also that there was likely disproportionation by selective fractionation of volatiles during uptake in the water. Quantification of the gas flux to the atmosphere by taking the ratio to the mass of the condensate sheen, although a useful 'first-guess' method, is thus very imprecise. Eventual estimates of condensate mass ranged from approximately 0.04 to 20 tonnes, over an affected area estimated from approximately 15 to 600 $km^2$. This wide range of estimates can potentially hamper a well-designed response effort (Ryerson et al., 2012). We emphasize the ability of the airborne chemical data to provide significantly more precise flow rate information than that provided by visual observations alone.

The evidence in the air plume for release of $CH_4$, $C_2$-$C_4$ alkanes, and benzene and toluene confirmed that the gas leak was not released from a significant depth. Initially it was not clear whether the gas leak was on the wellhead platform, or below sea-level, or both. After a Total press statement on 29 March 2012 and updated imagery on 30 March 2012, it became clear that there was indeed a gas leak at the wellhead on the platform. The airborne NMHC evidence supported the inference that release was indeed above sea level.

The height of the release was approximately the same as the aircraft sampling altitude in the lowest sampling cross-wind transects. Thus, the aircraft was able to fly through the core of the plume. This contrasts with the early situation in the BP Deepwater Horizon event, where release took place 1.5 km subsurface and $CH_4$ (Camilli et al., 2010;Yvon-Lewis et al., 2011), light alkanes, and light aromatics were essentially completely taken up in the water column (Reddy et al., 2012;Ryerson et al., 2012).





### 4.3 CH$_4$ isotopes

A further key goal of the airborne survey flights was to identify the geologic source of the gas leak using the CH$_4$ isotopic measurements ( $\delta^{13}$C$_{CH4}$) of the gas plume using the Tedlar bag and flask samples collected during the aircraft transects. This technique needs rapid sampling during the brief fly-through. Figure 8 shows $\delta^{13}$C$_{CH4}$ versus 1/CH$_4$ in air samples from the first two flights, following the Keeling plot methodology of (Pataki et al., 2003). The source gas has $\delta^{13}$C$_{CH4}$ of -42.3±0.7 ‰ (±2 σ) using geometric mean regression and a Monte Carlo style simulation to determine the propagation of errors into the fitting process where a geometric mean regression defines a line whose intercept on the $\delta^{13}$C$_{CH4}$ axis gives the end-member source value. The similarity of results from plotting separately the data from the two flights implies the gas source did not change between flights.

### 5 Discussion

### 5.1 Inference of the gas source

Compared to the first flight on 30 March 2012, the second flight on 3 April 2012 found significantly weaker plumes, suggesting that the gas source was depleting. This was significant in that it supported the inference that the source was comparatively small and depressurizing: i.e that the gas leak was indeed from a restricted source such as may be found in the Hod Formation and not from the main production depth (Bergerot, 2011). Information released by Total indicated that the main production depth had been plugged prior to the blowout.

The $\delta^{13}$C$_{CH4}$ isotopic ratio gives direct insight on the source of the gas. $\delta^{13}$C$_{CH4}$ is related to the fractionation that has occurred because of the thermal history of the geological source of the gas. In very hot deep gasfields, where early-formed biogenic gas may have escaped and later-formed gases include themogenic methane, CH$_4$ is typically enriched in $^{13}$C (i.e. $\delta^{13}$C$_{CH4}$ is less negative). In contrast, in shallower strata $\delta^{13}$C$_{CH4}$ is likely to be dominated by early-formed biogenic gas and lighter (i.e. more negative).

The source rocks below the main gasfield would have been at 5.5 km depth and at 200 $^{o}$C or more. In contrast, the over-pressured interval in the overlying Hod Formation is at about 4.2 km depth and 165 $^{o}$C (Isaksen, 2004). The gas in the Hod Formation likely formed *in situ*, trapped by the rock without early leakage of isotopically lighter gas. Thus gas in the Hod Formation will likely be much more negative in $\delta^{13}$C$_{CH4}$ than gas in the significantly hotter source regions underlying the Elgin field.

Methane isotopic information on the Elgin gasfield and the Hod Formation was not available; instead, we estimate these based on published stable isotopic values for ethane (C$_2$H$_6$) from the Elgin field. (Isaksen, 2004) show $\delta^{13}$C around -29 to -30 ‰. The data on the oils and ethane from the Elgin field suggest that hydrocarbons from the producing gas reservoir are in equilibrium with the setting (Isaksen, 2004). Under this assumption, and given the relatively high maturity of the field, in the Elgin production gas we expect $\delta^{13}$C in methane to be





similar to the $\delta^{13}$C ratio in ethane (Berner and Faber, 1996), perhaps in the range -25 ‰ to -35 ‰. If significant methane loss had occurred, or if methane had been introduced from below, we would expect it to be less negative. A $\delta^{13}C_{CH4}$ of -42‰ from whole-air samples collected from the gas plume is thus consistent with a source in the shallower, lower temperature Hod Formation, rather than the deeper main Elgin reservoir.

Alternatively a signature of 42‰ could be generated by mixing shallow gas with gas from the main reservoir. For future events, it is clear that the techniques described here combined with detailed isotopic analysis from the production field would considerably aid source identification.

**5.2 Dispersion modelling**

In order to assess any wider regional impact of the Elgin incident, HYSPLIT model simulations were carried for each day between 25th March 2012 at 18:00 UTC and 16th May 2012 at 18:00 UTC. For each day, a 72-hour dispersion forecast was produced and the concentration at 72 hours after initialisation was recorded. Then, a

time average of these 72-hour concentration distributions was produced. Thus, dispersion predictions were produced valid for the period 28th March 18:00 UTC until 19th May 2012 18:00 UTC. Calculations have only been made for $CH_4$ and assume that the methane is long-lived (lifetime much greater than the 3 day model runs). The source strength was allowed to vary temporally using an interpolated time series from the measured flow rate described earlier. Figure (9) shows HYSPLIT results broken down by week, integrated over all levels and

displayed over a domain containing all of Europe. The majority of the $CH_4$ was distributed mainly to the south of the source. Low concentrations of methane (<1 ppbV) travelled as far as mainland UK (principally the Humber Estuary, the North Norfolk Coast and the North Yorkshire coast) and continental Europe (Netherlands). The highest levels of concentration, however, appear to be confined to a rectangular box that extended from 56° N to 57° N and from 1° E to 3° E. This confinement is true at all levels. Above approximately 1 km above sea

level the concentrations were negligible. There is some evidence of the plume reaching as far South as Switzerland (at very low concentrations) during periods 28th March to 10th April and 9th to 19th May.

**6 Summary and conclusions**


These results demonstrate that a rapid-response airborne survey is able not only to quantify and track changes in the flux from the gas leak (e.g.(Conley et al., 2016;Ryerson et al., 2011;Ryerson et al., 2012)), but also to differentiate between potential source formations 4 to 5 km below ground and to provide accurate, independent, and time-critical information to guide operational response decisions. Moreover, the airborne measurement

provides an entirely external assessment, which is potentially useful to national regulatory and legal procedures. As in the *Deepwater Horizon* response, unavailability of reservoir compositional and isotopic data slowed interpretation, but in this case did not prevent the timely communication of robust and actionable results from these airborne survey flights.





Initially, a two-pronged approach was followed to resolve the Total Elgin event. Preparations were made to drill a relief well from outside the safety exclusion zone. This would have taken up to 180 days (Bellona, 2012). In parallel, an assessment was made of the safety of approaching the platform to control the well directly from the wellhead. As well as citing the flux estimates from this work, the Government Interest Group (2012) stated on

11 April that "Aerial surveys have been undertaken to obtain a qualitative assessment of the composition of the gas release, and modelling has been undertaken to investigate the dispersion of the release. The primary purpose of the modelling is to evaluate the explosion and safety risks." Permission for the successful dynamic kill was given on 3 May 2012. It is clear from the Government Interest Group statement that the FAAM aircraft results played an important role in the decision that it was safe to permit boarding the platform.

The cost of the two month shutdown of Elgin and connected fields was around UK £1, or roughly £15-20 million per day. Had the platform not been boarded, and the back-up plan for drilling a relief well been adopted instead, the shut-down could have lasted months longer, at much higher cost to the national fiscus. Given the statement of the Government Interest Group (2012) of the importance of the aircraft work in the safety

assessment, it is valid to assert that the FAAM aircraft measurements and the modelling they supported saved the UK Treasury a significant sum of potentially lost revenue had the shutdown lasted longer.

This study and earlier work by (Ryerson et al., 2011; Ryerson et al., 2012) and (Conley et al., 2016) work shows that airborne sampling can make important and very rapid findings to support decisive and effective response to

major atmospheric pollution events. In this case, fortunately, the gas leak, though serious, was relatively small and decreased with time. In addition to the Deepwater Horizon event discussed above, there are examples of other events where the effects have been more serious. In October 2015 blowout of a well connected to the Aliso Canyon underground storage facility in California resulted in a massive release of natural gas. Analysis of methane data from dozens of plume transects, collected during 13 research-aircraft flights between showed

atmospheric leak rates of up to 60 metric tons of methane per hour, an order of magnitude higher than the maximum leak rate calculated here from Elgin (Conley et al., 2016). From these measurements it was estimated that the amount of $CH_4$ released substantially impacted the State of California greenhouse gas emission targets for the year (California Environmental Protection Agency Air Resource Board, 2014) and was equivalent to the annual energy sector $CH_4$ emissions from medium-sized EU nations (EDGAR, 2016).


Therefore it is prudent to assume that there may be major future injections of unquantified emissions into the atmosphere from industrial activities, and that future pollution events may not be so forgiving. Moreover, other sources of gas releases to the atmosphere do occur, such as very large fires (Carvalho et al., 2011), or major volcanic emissions (see, e.g. Bluth et al. 1992; Sparks et al. 1997; USGS 2017). The methodology developed

here shows that independent airborne measurement can make major contributions to the management of such events and hence to public security.

**Acknowledgements**




The FAAM team is thanked for supporting the flights, including Maureen Smith, Steven Devereau, Doug Anderson, Guy Gratton, Steve Cowan, Angela Dean, Graeme Nott, Matt Gascoygne. For flight planning and resolving any safety issues staff from DirectFlight is thanked: Charlie Whittaker, Mark Robinson, Ian Ramsey-Rae, Robbie Voaden, Paul McCormick, Gaynor Ottoway, Peter Chappell, David Simpson, Mike Collins, and

5    Barbara Burge. Graham Dennis (Blacklocks Polo Books and Print) is thanked for very rapidly acquiring for us a copy of Turner (1994). Flights were supported by UK NERC/UKMO FAAM facility. The authors gratefully acknowledge the NOAA Air Resources Laboratory (ARL) for the provision of the HYSPLIT transport and dispersion model.



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



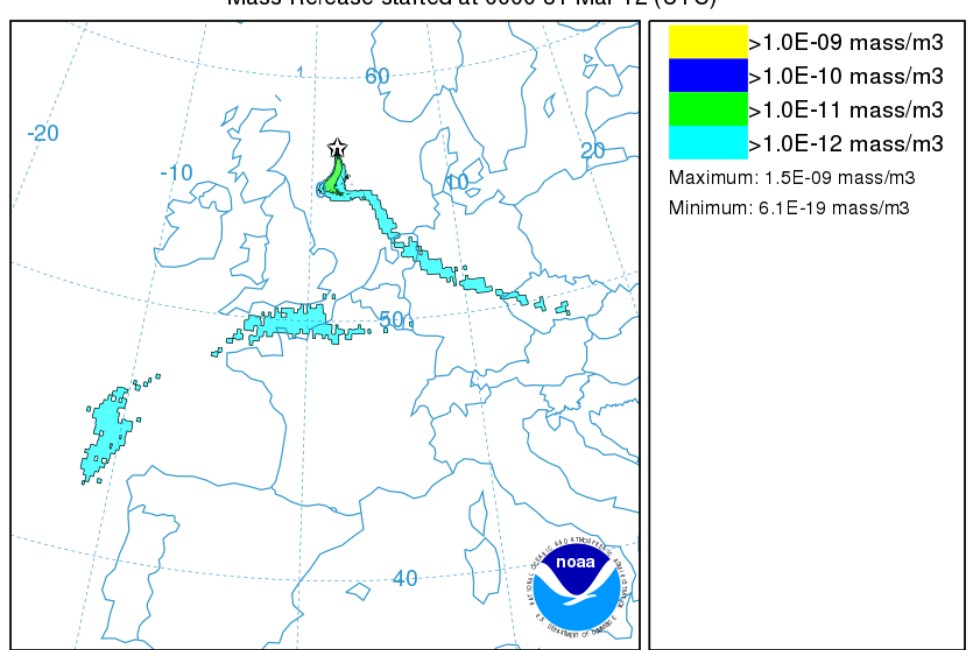

5    **Figure 2: Example of a prospective HYSPLIT model of the CH4 plume at 12.00 UTC, 02 April 2012. This assumed that the release rate was 23.5 kg s$^{-1}$ for the previous three days (see text for details).**





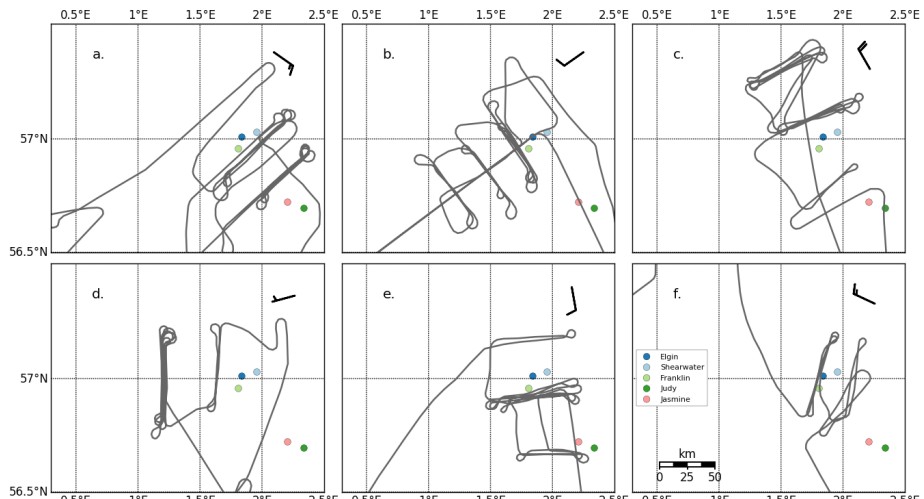

**Figure 3: Flight tracks for a.) B688 – 30<sup>th</sup> March 2012; b.)B689 – 3<sup>rd</sup> April 2012; c.) B690 – 17<sup>th</sup> April 2012; d.) B691 - 24<sup>th</sup> April 2012; e.) B693 – 4<sup>th</sup> May 2012; f.) B727 – 15<sup>th</sup> August 2012.**





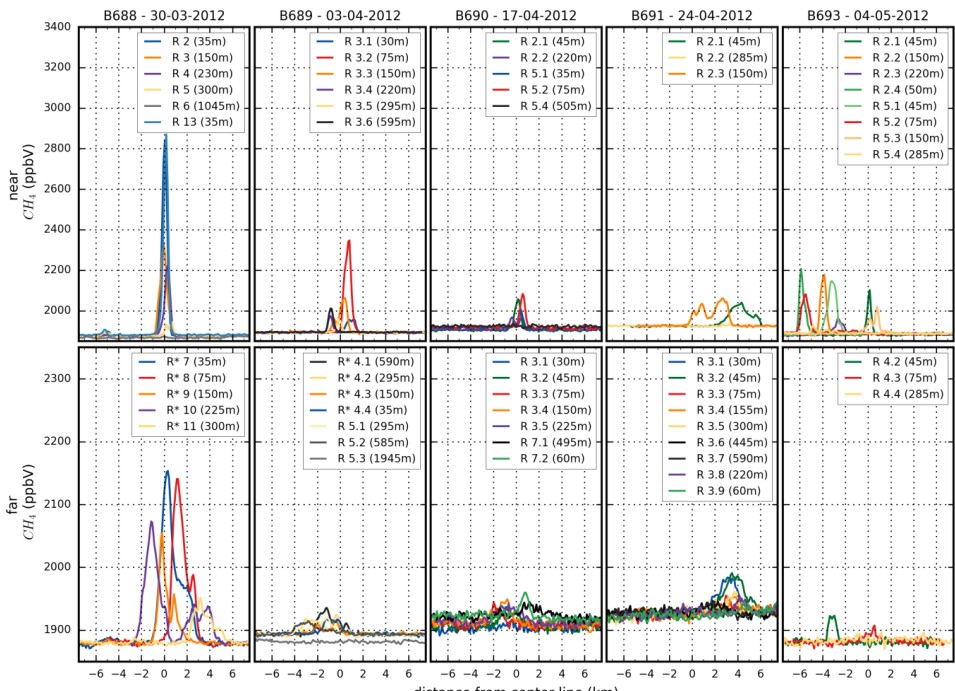

**Figure 4: CH4 measurements taken downwind of the Elgin rig during five flights. The upper panels show data taken at 5 NM and the lower panels data for 15 and 20 NM. Different colours show data for different runs. Runs at 15 NM downwind are denoted with an "*".**





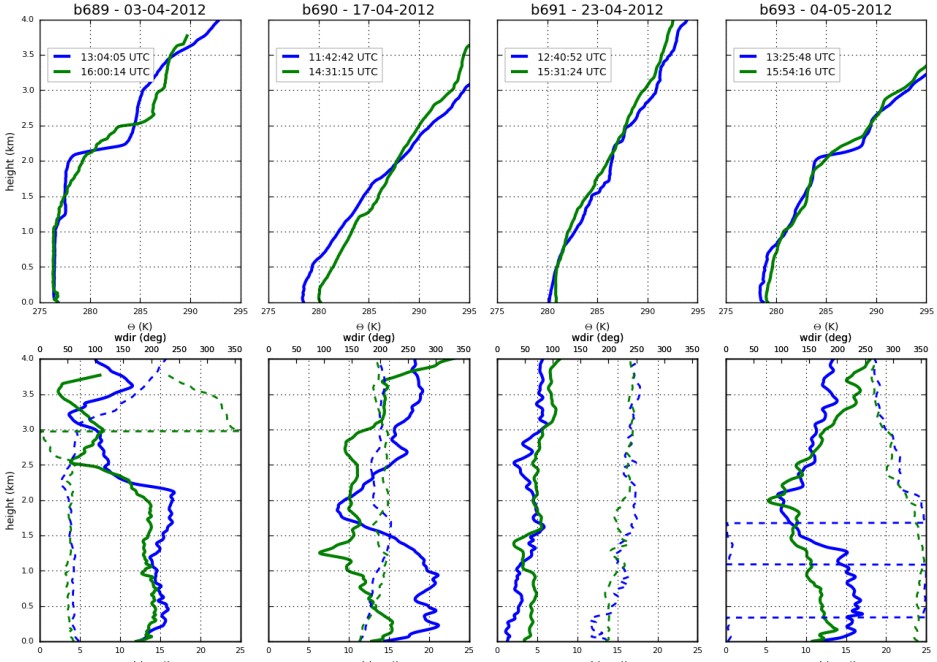

**Figure 5: Profiles of potential temperature (upper panel) and wind speed (solid lines) and direction (dashed lines) (lower panel) from dropsonde data early (blue) and late (green) in the flight for B689, B690, B691, and B693.**





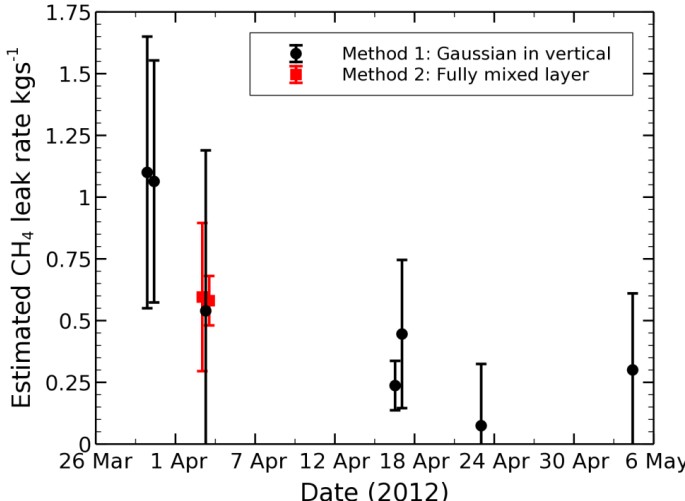

**Figure 6: Methane flow rate from flights on 30th March, 3rd April, 17th April, 24th April and 4th May 2012. The symbols in black show flow rates calculated using method 2 and those in red show flow rates calculated using method 1. Multiple results from the same flight are from different distances downwind from the Elgin rig and/or from different calculation methods. The time separation of multiple results from the same flight have been slightly exaggerated for clarity.**





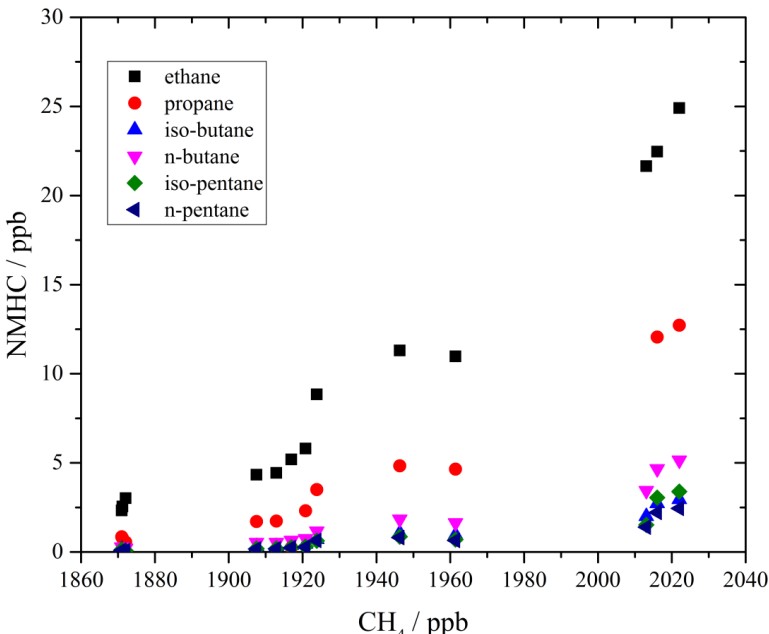

5    **Figure 7: NMHC and CH₄ relationship in Elgin plume samples. Data around 1860 ppb CH₄ represent typical background mixing ratios.**





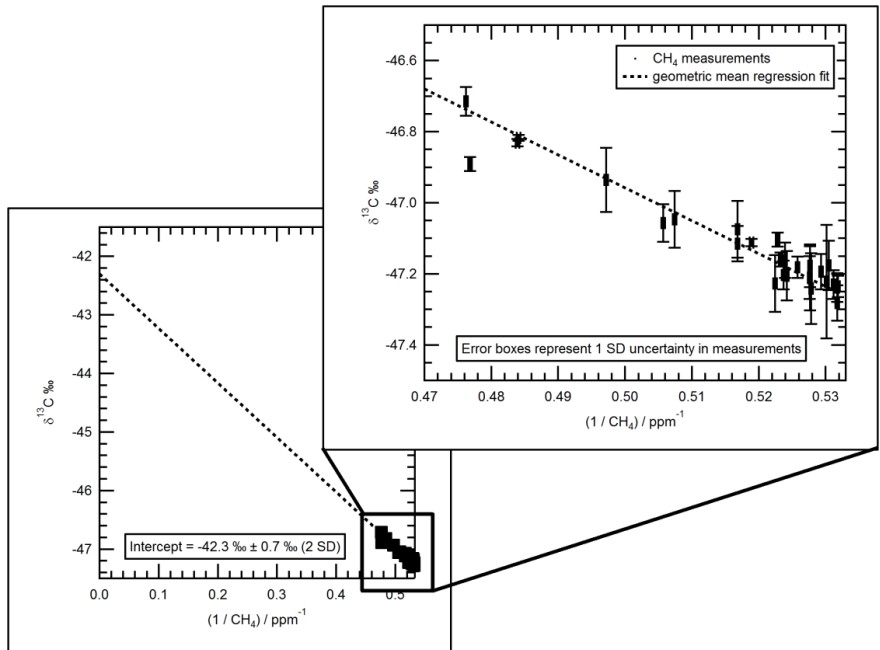

**Figure 8: Keeling plot of air samples. $\delta^{13}C_{CH4}$ -42.3±0.7 ‰. (2σ error: geometric mean regression).**





**Figure 9: Weekly HYSPLIT calculations of methane concentration over the European domain for weeks commencing (left to right, top to bottom) 28th March, 4th April, 11th April, 18th April, 25th April, 2nd May, 9th May and 16th May. Concentrations (colour shaded, ppb m) are vertically integrated from 0 to 2000 m; integrated concentrations < 1 ppb m not shown.**