# Peer review of "Flow rate and source reservoir identification from airborne chemical sampling of the uncontrolled Elgin platform gas release."

_Atmospheric Measurement Techniques, 2017_

## Referee Comment (RC1) · N. Cowern (Referee) · 15 Nov 2017

This is a significant and very well presented paper, clearly demonstrating the value of airborne plume sampling and analysis to guide decision making and rapid response to major hydrocarbon emission events.

One question that I do think still needs to be addressed in the paper is the apparently systematic variation in non-methane hydrocarbon ratios as a function of concentration, apparent in Fig. 7. For example, at a methane concentration of 1870 ppb the ethane/propane ratio appears to be around 5, while at 2020 ppb it is around 1.8. A similar, although weaker, trend occurs in the ratio between excess methane and ethane.

[Figure]

This does not quite justify the statement on page 13 lines 4-5 of the manuscript, that the ratios between methane and NMHCs were consistent. From the presented data the reader cannot see whether these concentration-related ratio changes are caused by time-dependent changes as the plume weakened, by spatial effects within the plume itself, or some other effect. As this clearly has significance for interpreting the results it should be discussed in the manuscript, and if necessary Figure 7 extended in order to clarify this. For example, different symbols should be used to identify data from different flights, in order to clarify any time dependence of plume composition.

---

## Referee Comment (RC2) · Anonymous Referee #2 · 22 Nov 2017

**1   Overview:**

Review of "*Flow rate and source reservoir identification from airborne chemical sampling of the uncontrolled Elgin platform gas release.*" by Lee *et al.*

Lee *et al.* present an analysis of the 2012 gas leak at the Elgin platform. The manuscript utilizes methane, NMHCs, and $\delta^{13}C_{CH_4}$ observations to determine the source of the leak (e.g., *"was it at the wellhead?"*, *"what formation is the leak coming from?"*). The authors claim to use two different methods to quantify the magnitude of the leak and argue that the methods compare well with each other. Based on the

abstract, I expected the authors to be comparing fundamentally different methods for quantifying the emissions. Instead the authors present an estimate of the source using a Gaussian plume and another estimate for a case when the plume has sufficiently mixed throughout the mixed layer. This is not necessarily problematic, but I don't think the methods used here can really be viewed as different. The manuscript is scientifically interesting but the description of the methods section is sloppy. I strongly suggest the authors go back through the derivations in Section 3 to ensure they are correct (e.g., Eq. 2 is clearly incorrect and the authors add/remove the "$(x, y)$" from equations seemingly at random between steps). Assuming this does not impact the actual analysis later on, the paper should ultimately be publishable because the findings are scientifically interesting. However, I suggest major revisions for the current manuscript.

**2   Major comments:**

**2.1   "Two" methods for estimating the source**

It seems that the authors are, in fact, just using one method for estimating the source. The authors are fitting a Gaussian plume based on their Eq. 1:

$$C(x, y, z) \;=\; \frac{q}{\pi \sigma_y \sigma_z U} \exp\left( -\frac{(y - y_0)^2}{2\sigma_y^2} - \frac{z^2}{2\sigma_z^2} \right).$$

The second method is simply a case where the plume has had time to sufficiently mix throughout the mixed layer, allowing them to neglect variations in the vertical and reduce Eq. 1 to (their Eq. 3):

$$C(x, y, z) \;=\; \frac{q}{\sqrt{2\pi}\sigma_y U H} \exp\left( -\frac{(y - y_0)^2}{2\sigma_y^2} \right).$$

As I mentioned above, this is not necessarily problematic but I do not think these should be presented as independent measures of the flow rate (unless I'm missing some fundamental difference). It gives the impression that multiple, independent methods for estimating the flow rate are in agreement. At best, it seems to indicate that the latter method (based on Ryerson *et al.* (2011)) is a valid simplification for when the plume is sufficiently mixed.

**2.2 Presentation of the methods**

The presentation of the methods is sloppy.

**2.2.1 Equation 2**

Eq. 2 is clearly incorrect. Eq. 1 is supposed to be a simplification of Eq. 2 where the plume has not reached the mixing height. However, the authors have written Eq. 2 as:

$$C(x,y,z) \;=\; \frac{q}{\pi\sigma_y\sigma_z U}\exp\left(-\frac{(y-y_0)^2}{2\sigma_y^2}\right)\left[\exp\left(\frac{z^2}{2\sigma_z^2}\right) + \exp\left(-\frac{(z+2H)^2}{2\sigma_z^2}\right) + \exp\left(-\frac{(z-2H}{2\sigma_z^2}\right.\right.$$

Taking $H \to \infty$ should return Eq. 1 but instead reduces to:

$$C(x,y,z) \;=\; \frac{q}{\pi\sigma_y\sigma_z U}\exp\left(-\frac{(y-y_0)^2}{2\sigma_y^2}\right)\left[\exp\left(\frac{z^2}{2\sigma_z^2}\right) + \exp\left(-\frac{(z+2H)^2}{2\sigma_z^2}\right) + \exp\left(-\frac{(z-2H}{2\sigma_z^2}\right.\right.$$

$$=\; \frac{q}{\pi\sigma_y\sigma_z U}\exp\left(-\frac{(y-y_0)^2}{2\sigma_y^2}\right)\left[\exp\left(\frac{z^2}{2\sigma_z^2}\right) + \exp\left(-\frac{(z+2\infty)^2}{2\sigma_z^2}\right) + \exp\left(-\frac{(z-2\infty}{2\sigma_z^2}\right.\right.$$

$$=\; \frac{q}{\pi\sigma_y\sigma_z U}\exp\left(-\frac{(y-y_0)^2}{2\sigma_y^2}\exp\left(\frac{z^2}{2\sigma_z^2}\right)\right)$$

I'm guessing the authors meant to write the following expression?

$$C(x,y,z) \;=\; \frac{q}{\pi\sigma_y\sigma_z U}\exp\left(-\frac{(y-y_0)^2}{2\sigma_y^2} - \frac{z^2}{2\sigma_z^2} - \frac{(z+2H)^2}{2\sigma_z^2} - \frac{(z-2H)^2}{2\sigma_z^2}\right)$$

**2.2.2 Nomenclature and the $x$, $y$, and $z$ dependence**

It was not initially clear where the $x$ dependence is coming from in Eq. 1 as $x$ does not appear anywhere on the RHS. $\sigma_y$ and $\sigma_z$ are both a function of $x$ and I'm used to computing these based on the atmospheric stability class (where the dependence of $\sigma$ on $x$ is clear because there's an expression showing it). It wasn't clear to me – until much later – that the authors were recomputing the dispersion parameters based on each transect (*that is what you are doing, right?*).

Related to this point, Eq. 4-7 are confusing because the authors include $(x)$, $(x,y)$, and $(x,z)$ for some of the equations but not others. For example, $C_0$ has the $(x)$ dependence in Eq. 5 but not Eq. 4 or when it is defined in Eq. 6. Is this the same $C_0$ in all of these cases? It would greatly help if the authors were consistent in their nomenclature (especially given the potential errors in the derivation of the earlier expressions...).

Additionally, the authors introduce an undefined $U'$ in Eq. 6 and an undefined $H'$ in Eq. 9, what are these variables? Are the primes typos?

**3   Minor comments:**

**3.1   What about the reflection at the ground in Eq. 1 (and subsequent equations)?**

I thought Gaussian plumes that included a vertical dependence typically included an imaginary source below the ground plane. This is because a plume cannot spread realistically in the vertical and will, instead, be reflected at the ground (imagine a cone that is sliced through the middle, that bottom half is reflected back up). This is why their Eq. 2 has those additional terms for a case where the plume has restricted mixing in the vertical. This would result in the dropping of the $1/2$ factor in the $z$ term in their Eq. 1:

$$C(x,y,z) \;=\; \frac{q}{\pi \sigma_y \sigma_z U} \mathrm{exp}\left(-\frac{(y-y_0)^2}{2\sigma_y^2} - \frac{(z-z_0)^2}{2\sigma_z^2} \underbrace{-\frac{(z+z_0)^2}{2\sigma_z^2}}_{\text{ground reflection}}\right),$$

for $z_0 = 0$ this would reduce to:

$$C(x,y,z) \;=\; \frac{q}{\pi \sigma_y \sigma_z U} \mathrm{exp}\left(-\frac{(y-y_0)^2}{2\sigma_y^2} - \frac{z^2}{\sigma_z^2}\right).$$

It does not seem that ground reflection is accounted for? It seems that this would impact the derived emissions?

**3.2   Figures**

The figures could be better.

- **Fig. 1:** Could use more description. Presumably the red box in the left panel is the domain of the right panel? Is the black dot the location of the platform?

- **Fig. 2:** The units on Fig. 2 are non-intuitive. Could the authors convert this to ppb? It also seems weird to have a massive NOAA logo. I can't think of any other paper where I've seen a large logo included in their figure. It doesn't seem appropriate for a publication. . .

- **Fig. 3:** I couldn't find a description of what "Shearwater", "Jasmine", "Judy", or "Franklin" were (a search of the manuscript didn't seem to show them anywhere except for the legend in Fig. 3), what are they? Other platforms?

- **Fig. 4:** Pretty hard to see what's going on in this figure, there's a lot of whitespace that's taken up by the legend (almost half of each panel is blank).

---

## Author Comment (AC2) · 4 Jan 2018

**Response to Reviewer 2 comments**

We thank the reviewer for their detailed comments and feel the changes we have made in response greatly improve the manuscript. Our responses to the individual points are detailed below.

**2.1. "It seems that the authors are, in fact, just using one method..."**

We take the reviewer's point, and we would go even further and say that both methods reflect the assumption that the concentration distribution is assumed to be of a Gaussian form. However, we would argue that the two methods of *solution* are so different that they warrant separate sections. In addition, the approach to the solution when there is no significant temperature inversion present ("Method 1") is novel, and alone justifies the separation into two "Methods". Taking into account the above, we have changed the text (extra text in red):

"Two different analysis approaches have been used, determined by the outcome of these measurements. They are referred to as Solution Method 1 and Solution Method 2 in this manuscript. Both solution methods reflect the assumption that the concentration distribution is assumed to be of a Gaussian form. However, the techniques of solution are different, and are here split into separate sections."

We have now changed the subsection headings to "Solution Method 1" and "Solution Method 2". We hope this now creates the distinction between the two *solutions*, while reflecting the fact that the solutions themselves refer to a single underlying assumption – the Gaussian assumption.

**2.2 Presentation of the methods**

We apologise for the poor presentation of the method and the equations. Hopefully the revised manuscript is an improvement. Specific changes to the manuscript bearing upon this point are described below.

**2.2.1. "Eq. 2 is clearly incorrect..."**

The reviewer is correct, this was a mistake in our original manuscript submission. The large round bracket at the end of our original eqn. (2) should have been used to terminate the first exponential term: this then corresponds to the reviewer's derivation and subsequent equation.

Our original equation (2):

$$\begin{split} \mathcal{C}(x,y,z) &= \frac{q}{2\pi\sigma_y\sigma_z U} exp\left(-\frac{(y-y_0)^2}{2\sigma_y^2} \left[\exp\left(\frac{z^2}{2\sigma_z^2}\right) + \exp\left(-\frac{(z+2H)^2}{2\sigma_z^2}\right) \right. \\ &\left. + \exp\left(-\frac{(z-2H)^2}{2\sigma_z^2}\right) \right] \right) \end{split}$$

Should have been:

$$C(x, y, z) = \frac{q}{2\pi\sigma_y\sigma_z U} exp\left(-\frac{(y-y_0)^2}{2\sigma_y^2}\right) \left[\exp\left(\frac{z^2}{2\sigma_z^2}\right) + \exp\left(-\frac{(z+2H)^2}{2\sigma_z^2}\right) + \exp\left(-\frac{(z-2H)^2}{2\sigma_z^2}\right)\right]$$

This has been corrected in the text. We are grateful to the reviewer for pointing this out.

**2.2.2 Nomenclature and the x, y and z dependence**

**" $\sigma_y$ and $\sigma_z$ are both functions x and I'm used to computing these based on the atmospheric stability class...it wasn't clear to me, until much later – that the authors were recomputing the dispersion parameters based on each transect (*that is what you are doing, right?*)"**

The reviewer is correct here. The dispersion parameters  $\sigma_y$  and  $\sigma_z$  are estimated from the measurements. Although attempts have been made (e.g. Song et al. 2003: reference below) to adopt Briggs-type formulas for use over sea surfaces, classified according to atmospheric stability, such ad-hoc approaches depend upon a simple manipulation of the land-based formulas (in the Song et al. example, a simple multiplication factor is used). This is *conceivably* appropriate for the lateral dispersion of gas but the vertical dispersion coefficient is unlikely to be represented well by this type of approach. The land-based formulas are derived from a large body of experiments: this is not the case for the marine equivalent (indeed, this would be a good subject for further study). Thus, we calculated the lateral and (when appropriate) vertical dispersion parameters based upon the aircraft measurements. This was perhaps not clear in the original manuscript, so we have added the extra text

"In land-based dispersion modelling, it is common to employ an approximation to the dispersion parameters  $\sigma_y$  and  $\sigma_z$ . (Examples may be found in Turner 1994.) These approximations (derived from many field experiments) are based upon the atmospheric stability and distance from source. Some attempts (e.g. Song et al. 2003) have been made to find similar approximations over sea surfaces; such attempts are not the result of field experiment, but rather of a manipulation of land-based formulae, and there is a question as to their validity. Thus, in the present study, we derive the dispersion parameters from the aircraft measurements, as described below."

This appears early in the manuscript to hopefully prevent the lack of clarity mentioned by the reviewer.

We have added the Song et al. reference to the manuscript.

Song, CH, et. al. (2003) "Dispersion and chemical evolution of ship plumes in the marine boundary layer: Investigation of  $O_3 / NO_y / HO_x$  chemistry". *Journal of Geophysical Research*, **108**, D4, 4143.

"Related to this point, Eq. 4-7 are confusing because... is this the same  $C_{\theta}$  in all these cases?.."

We have re-written Eq. 4-7 to show the full x, y, z dependence. Also we have added the extra text in the Solution Method 2 (formerly "Method 2") subsection: "N. B. the  $C_0$  here is different to the  $C_0$  for Solution Method 1." Hopefully this will alert readers to the different forms of the  $C_0$  term.

**"Additionally...undefined U'...undefined H'..."**

Apologies. These are typos and should be U and H respectively – now corrected.

**3.**

**Minor comments:**

**3.1 "What about the reflection at the ground..."**

We agree that reflections at the surface do need to be accounted for. Below is the relevant, full equation taken directly from Turner, *Workbook of Atmospheric Dispersion Estimates* (CRC Press, 1994; referenced in the text):

The notation used following  $\chi$  in parentheses is to give the three coordinates of the receptor location according to the coordinate scheme described above. Following a semicolon, the effective height of emission of the source is given.

The equation is given as four separate factors which are multiplied times each other. These four factors represent the dependency upon emissions, or the source factor, and what occurs in the three dimensions parallel to the three coordinate axes.

 $\chi(x,y,z;H) =$

Emissions factor

Downwind factor

1 \_\_\_\_\_

Q

Crosswind factor

| 1                        |     | y 2 |                 |   |
|--------------------------|-----|----------------|-----------------|---|
| $(2 \pi)^{1/2} \sigma_y$ | exp | _              | $2  \sigma_y^2$ | ] |

Vertical factor

$$\frac{1}{(2\pi)^{1/2} \sigma_{z}} \left\{ \exp\left[-\frac{(H-z)^{2}}{2\sigma_{z}^{2}}\right] + \exp\left[-\frac{(H+z)^{2}}{2\sigma_{z}^{2}}\right] \right\}$$

$$(2.1)$$

A brief explanation of the four terms follows.

1. The concentrations at the receptor are directly proportional to the emissions.

2. Parallel to the x axis, the concentrations are inversely proportional to wind speed as explained in Chapter 1.

3. Parallel to the y axis, that is, crosswind, the concentrations are inversely proportional to the crosswind spreading,  $\sigma_y$ , of the plume; the greater the downwind distance from the source, the greater the horizontal spreading,  $\sigma_y$ , the lower the concentration. The exponential involving the ratio of y to  $\sigma_y$  just corrects for how far off the center of the distribution the receptor is in terms of standard deviations. The receptor is y from the center since the crosswind distribution center is at y = 0, that is, directly above the x-axis.

4. Parallel to the z axis, that is, vertical, the concentrations are inversely proportional to the vertical spreading of the plume,  $\sigma_z$ ; the greater the downwind distance from the source, the greater the vertical dispersion and the lower the concentration. The sum of the two exponential terms in the vertical factor represent how far the receptor height, z, is

Chapter 2.

from the plume centerline in the vertical. The first term represents the direct distance, H - z, of the receptor from the plume centerline. The second term represents the eddy reflected distance of the receptor from the plume centerline, which is the distance from the centerline to the ground, H, plus the distance back up to the receptor, z, after eddy reflection.

After doing the multiplication the equation simplifies to:

$$\chi(\mathbf{x},\mathbf{y},\mathbf{z};\mathbf{H}) = \frac{Q}{2\pi \mathbf{u} \sigma_{\mathbf{y}} \sigma_{\mathbf{z}}} \exp\left[-\frac{\mathbf{y}^{2}}{2\sigma_{\mathbf{y}}^{2}}\right] \left\{ \exp\left[-\frac{(\mathbf{H}-\mathbf{z})^{2}}{2\sigma_{\mathbf{z}}^{2}}\right] + \exp\left[-\frac{(\mathbf{H}+\mathbf{z})^{2}}{2\sigma_{\mathbf{z}}^{2}}\right] \right\}$$
(2.1)

Note that the format of this equation includes an imaginary source (as mentioned by the reviewer); the last sentence in the text above refers to the eddy reflection. Thus, this form of the equation includes surface reflection.

In Turner's notation,  $\chi$  is the concentration,  $\sigma_y$  and  $\sigma_z$  are the dispersion parameters (we adopted the same notation), u is the ambient wind speed, H is the height of the source above the surface, and z is the height of the receptor. With a source at the surface (H = 0), Turner's (2.1) gives

$$\chi(x, y, z; 0) = \frac{Q}{2\pi u \sigma_y \sigma_z} \exp\left(-\frac{y^2}{2\sigma_y^2}\right) \left\{ \exp\left[-\frac{z^2}{2\sigma_z^2}\right] + \exp\left[-\frac{z^2}{2\sigma_z^2}\right] \right\}$$

Adopting our notation ( $C \equiv \chi$ ;  $U \equiv u$ ;  $q \equiv Q$ ) then we obtain

$$C(x, y, z; 0) = \frac{q}{\pi \sigma_y \sigma_z U} \exp\left(-\frac{y^2}{2\sigma_y^2}\right) \exp\left(-\frac{z^2}{2\sigma_z^2}\right)$$

Note that in Turner's formulation the y-axis constitutes the plume axis (i.e.,  $y_0 = 0$  in our notation). Allowing the possibility of a coordinate translation gives

$$C(x, y, z) = \frac{q}{\pi \sigma_y \sigma_z U} \exp\left(-\frac{(y - y_0)^2}{2\sigma_y^2} - \frac{z^2}{2\sigma_z^2}\right)$$

which is our equation (1).

We did not explicitly refer to reflections in the text - we apologise to the reviewer for any unnecessary confusion. We have now changed the text to: (extra text in red):

"The fundamental assumption is that the plume dispersion may be modelled by a Gaussian distribution. With the source at the surface, (z = 0), (see, e.g., equation 2.1 from Turner 1994):

$$C(x, y, z) = \frac{q}{\pi \sigma_y \sigma_z U} \exp\left(-\frac{(y - y_0)^2}{2\sigma_y^2} - \frac{z^2}{2\sigma_z^2}\right)$$
(Eq 1)

where *q* is the source strength (mass emission rate) of the methane leak, C(x,y,z) is the molar concentration which varies in the *x* (downwind), *y* (cross-wind) and *z* (vertical) directions and *U* is the mean prevailing wind speed. The  $\sigma^2_y$  and  $\sigma^2_z$  terms are the mean squared distances of the plume spread in the cross-wind and vertical directions (both growing by dispersion with down-wind distance). The source is fixed at *x* = 0. Note that this form of the equation includes reflection from the surface."

**Fig. 1: "could do with more description**" We have added the following text the figure caption:

The left map shows the location of the field in the North Sea, with the red rectangle shown on the right panel. The black dot indicates the location of the Elgin platform.

**Fig. 2. "The units on Fig. 2 are non-intuitive..."**

We have now changed this figure to be more visually appealing: the number of coloured contours has been increased and the NCAR logo has been removed. In addition, the contours now use PPB (converted via the molecular weight of CH4).

**Fig. 3: "couldn't find a description of what "Shearwater", "Jasmine", "Judy", or**

**"Franklin" were"**

We have added the following sentence to the figure caption:

The different platforms in the area (Elgin, Shearwater, Franklin, Judy and Jasmine) are shown by the different colour circles.

**Fig. 4: "Pretty hard to see what's going on in this figure, there's a lot of whitespace**

**that's taken up by the legend (almost half of each panel is blank)."**

We feel that we should leave this figure as it is. We wanted to plot all the runs on the same scale to allow the reader to easily see the changes in methane enhancement on the different flights and feel this is the best way to do this.

---

## Author Comment (AC1)

Response to Reviewer 1  comments

*We thank the reviewer for the very positive comments and for the suggestion on the section on the non-methane hydrocarbon section. We agree with the point about the ratios and have made the following changes and additions to the first paragraph of section 4.2 (indicated in coloured text):*

Non-methane hydrocarbons (NMHC) and other volatile organic compounds in the plumes were determined from whole-air flask samples by offline analysis (Hopkins et al., 2011; Lidster et al., 2014).  NMHC content was dominated by light alkanes ranging from >20 ppb ethane to <1 ppb benzene and <0.1 ppb higher monoaromatics.  A close relationship between elevated $CH_4$ and NMHCs (up to $C_5$) was observed in plume samples (Figure 7) with near consistent ratios.  It is noteworthy here that the NMHC ratios showed slight anti-correlation with methane mixing ratios.  For example, the ethane to propane ratio was found to vary from 2.7 - 5.5 down to around 2 at corresponding methane mixing ratios between 1871 and 2022 ppb. The relatively small number of observations makes it difficult to state with certainty whether the apparent relationship is indeed statistically significant or not.  What remains clear is that the absolute mixing ratios of methane and the lightweight NMHCs are well correlated. The plume was dominated by short chain (<$C_4C_6$) linear and branched chain alkanes and some larger monoaromatic compounds, with up to five alkyl groups substituents attached to the benzene ring.  No polycyclic aromatic compounds or oxygenated species were observed in any of the samples.

We have also changed figure 7 as suggested so it now shows which samples came from which flight.

---

## Author Response (AR2)

**Response to reviewer comments**

*rev. #1:*

*There is one pending issue (and 2 minor suggestions regarding Fig. 1):*

5 *\* Use of two methods and the presentation of the two methods:*

*I have no problem with the authors using slightly different derivations of the Gaussian plume. However, I think the authors need to rephrase the abstract because, as currently written, it strikes this reviewer as misleading. Based on a reading of the abstract I would assume the authors used two independent methods of estimating the flow rate and found them to be consistent: "two different*

10 *methods were used to calculate the flow rate: 1. Gaussian plume fitting in the vertical and 2. Direct integration of the plume. When both methods were used, they compared within 6% of each other". The abstract should be rephrased. I would suggest calling the methods something like "Gaussian plume" and "Gaussian plume (fully mixed)". This would also mean changing the legend in Figure 6 and the subsection titles in Section 3.*

We agree the abstract needs to be changed to reflect changes in the text and apologise for missing this in the first revision. Therefore we have replaced the sentence:

"Where appropriate, two different methods were used to calculate the flow rate: 1. Gaussian plume

20 fitting in the vertical and 2. Direct integration of the plume. When both methods were used, they compared within 6% of each other and within combined errors."

With:

25 **"The flow rate was calculated by assuming the plume may be modelled by a Gaussian distribution, with two different solution methods: Gaussian fitting in the vertical and fitting with a fully mixed layer. When both solution methods were used, they compared within 6% of each other, which was within combined errors."**

*\* Minor suggestions regarding Figure 1:*

*- There is a typo in Figure 1, the authors have misspelled "The" in the final sentence of the caption.*

Now corrected.

*- I also suggest that the authors add the other platforms in the region (Shearwater, Franklin, Judy, and Jasmine from Fig. 3) to Figure 1. This would help the reader get a feel for the region before going into the analysis. I incorrectly assumed that the Elgin platform was the only platform in the region when I was first going through the manuscript.*

Now added to the figure.

*rev. #2:*

*It is helpful to see in Fig. 7 of the revised manuscript which data originate from which flight. This makes it clear that there is a significant spatial dependence of the composition of the plume, which has not been acknowledged by the authors.*

*In lines 37-40 of the manuscript the authors note an anticorrelation between NMHC and methane mixing ratios but claim (without any quantitative basis) that this may not be statistically significant. In fact, it is quite clear that the mixing ratios of the heavier hydrocarbons (propane, butane and pentane) all fall systematically more rapidly than those of excess methane and ethane, over the entire range of methane mixing ratio. This is either a result of an annular composition profile of the plume, or (more likely) of deposition of the heavier fractions onto the cold water surface along the length of the plume. This is so obvious that some comment is required in order to properly characterise the results.*

*Acknowledging this has no implications for the flow rate estimates in the paper, which are based on the methane data alone. However, it would not be proper to ignore the fact that the physical situation is more complex than the simple model description used in the paper.*

We agree that this needs a little more explanation and we thank the reviewer for their suggestion. So in section 4.2 we have replaced the sentences:

"A close relationship between elevated CH4 and NMHCs (up to C5) was observed in plume samples (Figure 7) with near consistent ratios. It is noteworthy here that the NMHC ratios showed slight anti-correlation with methane mixing ratios.  For example, the ethane to propane ratio was found to vary from 2.7 - 5.5 down to around 2 at corresponding methane mixing ratios between 1871 and 2022 ppb. The relatively small number of observations makes it difficult to state with certainty whether the apparent relationship is indeed statistically significant or not. What remains clear is that the absolute mixing ratios of methane and the lightweight NMHCs are well-correlated. "

With

NMHCs up to $C_5$ all showed enhancements corresponding to enhanced $CH_4$. However it is noteworthy that the mixing ratios of the heavier hydrocarbons (propane, butanes and pentanes) all fall systematically more rapidly than those of excess methane and ethane, over the entire range of methane mixing ratio (as shown in figure 7). We believe this is likely caused by the heavier weight compounds condensing more readily to the cold water surface along the length of the plume due to their increased solubility.

**Flow rate and source reservoir identification from airborne chemical sampling of the uncontrolled Elgin platform gas release.**

James D. Lee[1*], Stephen D. Mobbs[2], Axel Wellpott[3], Grant Allen[4], Stephane. J-B. Bauguitte[5], Ralph R. Burton[2], Richard Camilli[5], Hugh Coe[4], Rebecca E. Fisher[6], James L. France[6,7], Martin Gallagher[4], James R. Hopkins[1], Mathias Lanoiselle[6], Alastair C. Lewis[1], David Lowry[6], Euan G. Nisbet[6], Ruth M. Purvis[1], Sebastian O'Shea[3], John A. Pyle[8], Thomas B. Ryerson[9].

[1] National Centre for Atmospheric Science, University of York, York, UK.
[2] National Centre for Atmospheric Science, University of Leeds, Leeds, UK.
[3] Facility for Airborne Atmospheric Measurements, Cranfield University, Bedford, UK.
[4] School of Earth, Atmospheric and Environmental Sciences, University of Manchester, Manchester, UK,
[5] Department of Applied Ocean Physics and Engineering, Woods Hole Oceanographic Institution, Woods Hole, MA, USA.
[6] Department of Earth Sciences, Royal Holloway University of London, Egham, TW20 0EX, UK.
[7] School of Environmental Sciences, University of East Anglia, Norwich, UK.
[8] National centre for Atmospheric Science, Department of Chemistry, University of Cambridge, Cambridge, UK
[9] Chemical Sciences Division, Earth System Research Laboratory, NOAA, Boulder, CO, USA.

*Corresponding author, email: james.lee@york.ac.uk

**Abstract.** An uncontrolled gas leak from 25 March to 16 May 2012 led to evacuation of the Total *Elgin* well head and neighbouring drilling and production platforms in the UK North Sea. Initially the atmospheric flow rate of leaking gas and condensate was very poorly known, hampering environmental assessment and well control efforts. Six flights by the UK FAAM chemically-instrumented BAe-146 research aircraft, were used to quantify the flow rate. The flow rate was calculated by assuming the plume may be modelled by a Gaussian distribution, with two different solution methods: Gaussian fitting in the vertical and fitting with a fully mixed layer. When both solution 
[revised manuscript text omitted]